# Fungal Melanin and the Mammalian Immune System

**DOI:** 10.3390/jof7040264

**Published:** 2021-03-31

**Authors:** Sichen Liu, Sirida Youngchim, Daniel Zamith-Miranda, Joshua D. Nosanchuk

**Affiliations:** 1Department of Medicine, Albert Einstein College of Medicine, Bronx, NY 10461, USA; siliu@montefiore.org (S.L.); daniel.zamithmiranda@einsteinmed.org (D.Z.-M.); 2Department of Microbiology, Faculty of Medicine, Chiang Mai University, Chiang Mai 50200, Thailand; syoungchim@gmail.com; 3Department of Microbiology and Immunology, Albert Einstein College of Medicine, Bronx, NY 10461, USA

**Keywords:** melanin, fungus, yeast, immune response

## Abstract

Melanins are ubiquitous complex polymers that are commonly known in humans to cause pigmentation of our skin. Melanins are also present in bacteria, fungi, and helminths. In this review, we will describe the diverse interactions of fungal melanin with the mammalian immune system. We will particularly focus on *Cryptococcus neoformans* and also discuss other major melanotic pathogenic fungi. Melanin interacts with the immune system through diverse pathways, reducing the effectiveness of phagocytic cells, binding effector molecules and antifungals, and modifying complement and antibody responses.

## 1. Introduction

Melanins are a family of structurally complex dark pigment polymer present in all biological kingdoms [1]. The polymer is made up of covalently linked indoles, but the overall structure is granular; however, detailed structures are not well characterized because it is a collection of polymers with mixed pre-indole structures [2]. Moreover, melanins are amorphous and are not suitable for study by crystallography or cryo-electron microscopy, which has led to complicated efforts to solve the structure of these natural pigments. Mammalian melanin biogenesis happens by oxidation of L-tyrosine via tyrosinase in melanocytes, which are neural crest-derived dendritic cells [3,4]. However, melanins in bacteria, fungi, and helminths are produced through the polyketide synthase (PKS) pathway or catalyzed by phenoloxidase [5]. This review will focus on melanin in fungi and its function in pathogen–host interaction with a particular emphasis on the immune system.

## 2. Melanin Synthesis

Fungi synthesize melanin via two main pathways, namely 1,8-dihydroxynaphthalene (DHN) and l-3,4-dihyroxyphenylalanine (L-DOPA) (Figure 1 and Table 1). In the DHN pathway, 1,3,6,8-tetrahydroxynaphthalene (1,3,6,8-THN) is first synthesized through PKS, which is a multi-domain enzyme complex that produces polyketides. Polyketides are products derived from acetyl-CoA or propionyl-CoA with malonyl-CoA or methylmalonyl-CoA. The condensation reactions are driven by decarboxylation, yielding a beta-keto functional group [6]. This is then followed by reduction and dehydration reactions that eventually produce DHN. It is the polymerization of DHN that leads to the formation of melanin [5,7,8]. Polyketides are a class of secondary metabolites mainly produced in bacteria, fungi, and plants, and they serve very different purposes in our society [9]. Polyketides such as macrolide, tetracycline, and amphotericin antimicrobials serve tremendous value, while aflatoxin can be lethal to mammals [10,11].

The L-DOPA pathway is similar to mammalian melanin biosynthesis, where the pathway typically uses either L-DOPA or tyrosine as starting molecules. If the pathway starts with tyrosine, tyrosinase will perform a two-step oxidation, turning tyrosine into dopaquinone. Similarly, laccase is the enzyme responsible for converting L-DOPA into dopaquinone. Dopaquinone is then turned into leucodopachrome (cyclodopa) and then oxidized to dopachrome. Dopachrome then goes through tautomerization to form dihydroxyindoles, which are simultaneously oxidized and polymerized to produce DOPA melanin [8].

## 3. *Cryptococcus neoformans*

*C. neoformans* is one of the most well studied pathogens for melanization (Figure 2). *Cryptococcus neoformans* is unique among pathogenic fungi as it solely relies on the L-DOPA pathway and requires exogenous phenolic substrates to form melanin. After the polysaccharide capsule, melanin is the second most important virulence factor in *C. neoformans* as it has been calculated as contributing to 14% of the pathogen’s total virulence [16]. This notable aspect of cryptococcal melanization has led to extensive study of this polymer in *Cryptococcus* spp. [17]. Other species can produce melanin using endogenous compounds or exogenous substrates, and some can produce more than one type of melanin [12,18,19].

In the early 1980s, Kwon-Chung et al. produced melanin-deficient strains (*Mel^−^*) via UV irradiation and observed that the mutants lacked virulence as the cells were cleared from mouse organs, whereas wild-type (WT) cells expanded in numbers, especially in the brain [21]. The same study showed that *Mel^−^* mutants were defective in the active transport system for diphenolic compounds and phenoloxidase, which is the first enzyme needed in the L-DOPA melanogenesis pathway. A subsequent study confirmed that the loss of phenoloxidase activity was responsible for the *Mel**^−^* mutant phenotype [22]. Williamson next discovered that laccase, a phenoloxidase, was encoded by the lac1 gene [23]. Laccase was linked to *C. neoformans* virulence in vivo through disruption of the 5′ end of the *lac1* gene [24]. Interestingly, *C. neoformans* has two laccase genes, where *lac2* is 75% similar to lac1, but basal transcript levels of *lac2* are much lower, and mutation of the gene only induced a mild delay in melanin formation [25].

### 3.1. Melanosomes and Melanin Location

Melanin synthesis needs to occur in a contained environment as its intermediates are reactive to other surrounding substances [26]. Melanin is synthesized in melanosomes within human melanocytes. In *C. neoformans*, melanization occurs in specialized vesicles that are also called melanosomes [27]. In *C. neoformans*, laccase was found to be one of many products packaged into vesicles that could be trapped in the cell wall or secreted to the extracellular milieu [28]. Indeed, RNA interference (RNAi) of sec6, which encodes a protein required for an event leading to fusion of post-Golgi vesicles with the plasma membrane, prevented melanization by blocking vesicle access [29]. Internal “melanosomes” are also observed in *C. albicans*, *Cladosporium carrionii*, and *Hormoconis resinae* [14,30]. Work in *Aspergillus* has revealed that disruption of endosomal sorting blocks the deposition of melanin in the cell wall, further supporting non-conventional secretory mechanisms driving proper polymer formation and draws marked similarities to mammalian melanosome biology [31]. The existence of cryptococcal melanosomes explains how melanin is distributed in the cell wall, with the melanosomes depositing diffusely in ring-like layers in the cell wall [19,32]. However, melanin’s distribution is pathogen-dependent. Whereas *C. neoformans* has melanin located in the innermost layer of the cell wall [33], melanin is located on the surface or outer wall layer in *Candida albicans*, *Aspergillus* spp., *Paracoccidioides* spp., *Coccidioides* spp., *Histoplasma capsulatum*, and *Sporothrix schenckii* [13,34,35,36,37,38].

### 3.2. Melanin and Cell Wall

Melanin’s role in the cell wall has been extensively studied in *C. neoformans*. Many of these studies utilized melanin “ghosts”, which are melanin spheres left over from enzymatic digestion and chemical breakdown of the cell wall. Ghosts formed in the presence of ^13^ C showed that melanin is an integrated part of the cell wall, forming covalent bonds with polysaccharides within the cell wall [26]. *C. neoformans* melanin ghosts are made of two to five concentric layers with thicknesses ranging from 50 to 75 nm per layer. The layers become thicker as cultures become older [39]. The presence of melanin impacts the cell wall’s properties. For example, a comparison of WT melanized and non-melanized *A. fumigatus* mutants showed that the conidial wall of the reference strain was composed of several superimposed layers with a thick electron transparent inner layer and two thin electron-dense outer layers, while mutant conidia showed a cell wall devoid of the outermost layer. Scanning electron microscopy (SEM) observed that pigment-less mutants have smooth-walled conidia without ornamentation, and further analysis showed that the mutant cell wall was also less electronegative and hydrophobic [40]. Interestingly, there are still pores present in the cell wall after melanization, which permit the transport of molecules across this barrier. In *C. neoformans*, pore sizes were markedly reduced in pigmented cells as pore radii were estimated to be 4 and 10.6 nm for melanized and non-melanized types, respectively [41]. *C. neoformans* cell wall pore size also correlated with the age of the culture, as day 4 melanin ghosts displayed the greatest porosity at 41 μL Å^−1^ g^−1^ at 16.4 Å, while day 10 melanin ghosts’ porosity was 15 μL Å^−1^ g^−1^ at 10 Å. The same study also showed that an antibody against melanin decreased the total porosity from 16 to 8 μL Å^−1^ g^−1^ [39], which has important implications for drug targeting.

In terms of its structure within the cell wall, *C. neoformans* melanin is described as being composed of irregular granules 50–80 nm in diameter, and similar granular structures are observed in other fungi such as *Fonsecaea pedrosoi*, *Hortaea werneckii*, and *Agaricus bisporumand* [39,42,43,44]. In *C. neoformans*, these granules are cross-linked with chitin, which is a polymer of β(1,4)-linked N-acetylglucosamine (GlcNAc) subunits joined in antiparallel chains by hydrogen bonding to produce strong microfibrils [45]. Chitin is synthesized by chitin synthase (CHS), which uses UDP-N-GlcNAc as a substrate to grow the polymer [32].

Through *chs*-deficient mutants, we were able to learn how chitin interacts with melanin in *Wangiella dermatitidis*, *C. neoformans*, and *Candida albicans*. In *W. dermatitidis*, the WdCHS4 knockout mutant allowed us to observe a decrease in melanin deposition within the cell wall concomitant with an increase in the pigment’s concentration extracellularly. In *C. neoformans*, a similar observation occurred in *cda1*, *cda2*, and *cda3* deletion mutants. These genes are responsible for the deacetylation of chitin, producing chitosan. Chitosan not only helps in cell integrity and bud separation, but the disruption of these genes resulted in a “leaky melanin” phenotype that exhibited melanin suspended in the supernatant, while still retaining some melanin within the cell [46]. The *chs3* and *csr2* genes, which are a chitin synthase and a regulator of chitin synthase, respectively, were also found to be necessary to retain melanin within the cell as their mutants also demonstrated the “leaky” phenotype [47,48]. *C. neoformans* cda1∆2Δ3Δ mutants have decreased virulence, and mice inoculated with this mutant generate a strong inflammatory response followed by enrichment of Th1-type T cells in lung tissue, producing immunity against the WT strain [49].

*C. albicans* did not exhibit these leaky phenotypes but displayed repressed melanin externalization, and seemingly different structural types of chitin have different melanin localization effects. The chitin synthase (*chs2*)-deficient mutants showed accumulation of melanin grains and melanosomes within the cells, while *chs8*- and *chs8*/*chs2*-deficient mutants displayed WT behavior [14]. Similarly, melanin externalization was impaired with disruptions in the chitinase genes CHT2 and CHT3 or the chitin pathway regulator ECM22.

### 3.3. Survival Advantage

Melanin production enhances fungal survival in the environment. Facultative melanotic *C. neoformans* produces melanin while dwelling in the soil [50]. The polymer absorbs electromagnetic radiations such as UV and gamma radiation, much like its role in humans, and allows fungi to survive in temperature extremes [51,52]. It is also partially responsible for the survival of mold inside nuclear reactors [53,54]. Melanin not only protects against toxic radiation, but the polymer also enables fungal cells to harvest high-energy radiation for growth [55,56], which means that fungi are limited autotrophs. The primary mechanism for melanins being so efficient with shielding from biotic and abiotic factors relies not only on its extremely stable structure but also on the fact that it has a remarkably broad optical absorption, paramagnetism, and charge transport. These physico-chemical properties are since, as polyphenols, melanins are polymers composed by series of aromatic rings, thus allowing electron resonance and also mediating energy transfer reactions in the cells [57].

Cells with melanin are structurally stronger than non-pigmented cells as melanized fungi are more resistant to enzymatic degradation [58]. The carboxyl, phenolic, hydroxyl, and amine groups within the polymer compound are perfect sites for heavy metal binding [59], and the ability of melanized *C. neoformans* to resist silver compounds is linked to its binding capacity [60].

Melanin alters antimicrobial pharmacokinetics. In vitro efficacies of various antibiotics (aminoglycosides, tetracyclines, and vancomycin) and antifungals (amphotericin B and caspofungin) are significantly altered in the presence of melanin [61,62,63]. In fact, according to Barza et al., tobramycin’s efficacy decreased by 80% in the presence of melanin [62]. Binding, but no efficacy reduction, is seen with fluoroquinolones, penicillins, cephalosporins, and azoles [61,63].

### 3.4. Melanin and Host Effector Cells

Melanin is a negatively charged polymer [64]. Melanization increased the overall cellular negative charge by 3 to 33% in nine different encapsulated strains of *C. neoformans* and by 86% in an acapsular strain [65]. It has been hypothesized that cell charge might affect phagocytosis. Indeed, phagocytosis of *C. neoformans* can be affected by the surface hydrophobicity and charge [66]. Microbial cell charge has been shown to affect rates of phagocytosis by neutrophils and monocytes [67,68]. In fact, melanization of *C. neoformans* interferes with phagocytosis in vivo [69].

The mere presence of melanin, regardless of charge, improves pathogen survival. Melanized and non-melanized mutants were incubated with capsule-binding monoclonal Mab 2H1 antibodies and murine macrophages. Melanized cells conferred a 31% survival advantage over the mutant with 98.5% of the cells surviving [70]. The authors attributed the resistance to phagocytosis to melanin’s resistance to reactive oxygen species (ROS), but later research suggests that more complicated mechanisms are at play, and some seem to be pathogen-specific. Melanin’s negative charge and capacity to bind diverse molecules make cryptococcal cells less susceptible to cationic antimicrobial peptides released by phagocytes, such as defensins, than non-melanized cells [71]. *C. neoformans* is able to escape macrophages via nonlytic exocytosis [72]. Remarkably, laccase expression regulates both melanin formation and rates of nonlytic exocytosis [73].

Melanin-producing strains of *Cryptococcus* spp. suppress the host immune and inflammatory responses [74,75]. Rosas et al. induced granuloma formation in mouse peritoneal cavity after injecting isolated *C. neoformans* melanin particles. Granulomas were seen in the liver, spleen, and lung. Granuloma formation around melanin particles resembled a foreign body reaction [76]. This might be due to the immune system’s inability to break down foreign melanin. In chronic and latent cryptococcal infections, the yeast cells are usually encased in granulomas, similarly to those observed with melanin particles [77]. Interestingly, fungal pathogens that can induce latency and granuloma formation produce melanin [74]. This suggests that melanin might have immunomodulatory effects, preventing the pathogens’ clearance. In fact, there is early evidence that laccase protects *C. neoformans* from alveolar macrophages antifungal activities [78].

Since granulomas contain a wide range of macrophage morphologies, there have been numerous investigations focused on macrophages [79]. Indeed, Tajima et al. found that solubilized melanin suppresses macrophage function [80]. Solubilized melanin did not affect the survival of the macrophages, but, interestingly, induced proliferation at a very high concentration, 100 µg·mL^−1^. The macrophage’s phagocytosis ability was attenuated significantly, along with ROS production. Regarding cytokine production, TNF-a, IL-1b, and IL-6 were suppressed [80]. Interestingly, another study did not find differences in IL-2, IL-10, TNF-α, and IFN-γ levels, while there was twice the amount of IL-4 and an elevated monocyte chemoattractant protein-1 (MCP-1) level in lung tissues infected with melanized versus non-melanized *C. neoformans* [69].

Not only does melanization influence macrophage activities, but melanin also modulates T cell responses. *C. neoformans* strains 52 and 145 have opposite inflammatory reactions in experimental mouse models and these strains have been used to dissect out adaptive immune processes in cryptococcosis. The two strains differ in that strain 145 produces more melanin and strain 52 produces a stronger pulmonary inflammatory response [81]. In a pulmonary infection model, strain 145 had delayed responses from neutrophils and macrophages and a diminished lymphocyte response, especially CD4+ T cells, compared to strain 52. The lack of CD4+ T cell proliferation was also seen in peripheral lymph nodes [82]. Accordingly, alveolar macrophages produced 70% less TNF-α when they were incubated with the high melanin-producing strain 145 versus strain 52. These changes are also present during murine central nervous system infections. The high melanin-producing strain suppressed IL-12, IL-1β, TNF-α, IFN-γ, and inducible nitric oxide synthase (iNOS) to almost non-existent levels and resulted in a 100% mortality rate [83].

### 3.5. Antibody

Immunization of mice with cryptococcal melanin induces a robust antibody response with significant increases in IgM and IgG levels [84]. Moreover, antibodies to melanin naturally occur during cryptococcosis [85]. The development of cryptococcal melanin-binding monoclonal antibodies (mAbs) accelerated the study of melanin in pathogenic fungi [20], particularly as the mAbs reacted with diverse biological and synthetic melanins [35]. The melanin-binding mAbs revealed a new mechanism for antibody activity in which the mAbs interfere with properties of the polymer to inhibit growth [86]. In fact, mAbs block pores in the layered melanin [39], which likely interferes with the transport of macromolecules. Melanin-binding mAbs also prolonged the survival of mice lethally infected with *C. neoformans* [86].

### 3.6. Complement System

The complement system is a major effector arm of host defense. Complement C3 fragments are deposited in the *C. neoformans* capsule and enhance the clearance of these cells by phagocytes [87,88]. Interestingly, C3 also deposits on cryptococcal melanin ghosts, and this process is due to activation of the alternate complement pathway [76]. Deposition of complement fragments also occurred with ghosts injected into the lungs of mice. Notably, the presence of melanin did not affect the kinetics of C3 binding to the cryptococcal capsule. However, the C3-labeled melanin ghosts induced the formation of small granulomas in mouse lungs.

## 4. *Aspergillus fumigatus*

*A. fumigatus* is another pathogenic fungus that undergoes melanization, and pigment formation in this species has been the subject of extensive investigation. 1,8-dihydroxynaphthalene (DHN) melanin is the major melanin found in *A. fumigatus* [89]. A set of six genes are needed for melanin synthesis: *pksP*, *ayg1*, *arp2, arp1*, *arb1*, and *arb2*, ordered from upstream to downstream [90]. Melanin from *A. fumigatus* has many similarities to melanin from *C. neoformans*, as both provide protection from ultraviolet light and scavenge ROS generated by phagocytes [91]. *Albino conidia* incubated with phagocytes induced a 10-fold increase in ROS production compared with WT, which suggests that the melanin increased ROS scavenging abilities in the WT conidia. Moreover, the albino mutants were more effectively killed by monocytes than wild-types [92].

Melanin regulates host pro-inflammatory cytokine responses by physically masking fungal pathogen-associated molecular patterns (PAMPs) from immune recognition such as that observed by the rodlets layer. Chai et al. demonstrated that albino conidia were able to generate much higher IL-6, IL-10, and TNF-α levels compared to WT. IL-6 and IL-10 had roughly 12-fold and 5-fold higher levels, respectively. The authors pinpointed that albino conidia had β-glucan and other PAMPs such as mannans more readily available to bind to dectin-1, Toll-like receptor 4 (TLR4) and Mannose receptors on peripheral blood mononuclear cells (PBMCs) [93]. As conidia mature, they swell and germinate, and this process exposes β-1,3-glucan on their surface, which induces an immune response [94].

Jahn et al. observed that ΔpksP melanin-deficient conidia were more effectively phagocytosed and killed by macrophages when compared with WT conidia [95]. Thywissen et al. next demonstrated that the albino ΔpksP mutant conidia had a 3.5-fold increase in vacuolar-type ATPase-dependent phagosomal acidification observed in alveolar and monocyte-derived macrophages (~20% acidified conidia vs. 70%). There were virtually no acidified phagosomes containing WT conidia in granulocytes, but ~50% in the mutant group. The pH within the mutant-containing phagolysosome was 5, while the WT had a pH of 6, and the lower pH results in more effective actions by phagosomal enzymes. Interestingly, synthetic DOPA melanin did not prevent acidification, only DHN melanin did [96]. However, among *Aspergillus* spp., *A. flavus* was the most effective in suppressing acidification, closely followed by *A. fumigatus*. Microtubule-associated protein 1A/1B-light chain 3 (LC3)-associated phagocytosis (LAP) is a non-canonical autophagy pathway that is linked to certain pattern recognition receptors that trigger phagosome formation [97]. *A. fumigatus* melanin inhibits calcium-calmodulin signaling on the protein Rubicon, a key regulator in the LAP pathway [98,99], and Rubicon directly interacts with the p22phox subunit and facilitates NADPH oxidase activation during phagocytosis [100]. Melanin ultimately blocks the p22phox NADPH oxidase subunit from localizing on the phagosome membrane, thus blocking the assembly of the oxidase complex. Melanin’s effect on the NADPH oxidase complex is conserved in *A. nidulans* as well [101].

The effects of melanin vary on different immune cells. Dendritic cells (DCs) are not stimulated by *A. fumigatus* melanin. Bayry et al. demonstrated that DCs failed to produce cytokines TNF-α, IL-1β, IL-6, and IL-10 in response to melanized *A. fumigatus* conidia, and DCs treated with WT melanin ghost also failed to activate T cells [102]. The group interestingly found that Δ*pksP*, Δ*ayg1*, and Δ*arp2* mutants increased different amounts of acetyl-CoA, malonyl-CoA, and 1,3,6,8-THN, respectively. These mutants displayed altered cell walls with unmasked surface structures and were able to activate DCs.

Mammals have evolved various systems to combat melanized fungal pathogens. On the surface of mouse endothelial cells, there is a melanin-sensing C-type lectin receptor (MelLec) that recognizes DHN melanin in the conidial spores of *A. fumigatus* and other DHN-melanized fungi, such as *Cladosporium cladosporioides* and *Fonsecaea pedrosoi* [103]. The expression of this C-lectin is essential for protection against disseminated aspergillosis, and albino mutants are not recognized by the receptor. Macrophages change their metabolism in the presence of melanin, especially glycolysis metabolism, which is required for defense against *Aspergillus*. Goncalves et al. elucidated that DHN melanin blocks endoplasmic reticulum calcium/calmodulin signaling, which activates glycolysis and mammalian target of rapamycin (mTOR)-mediated defense against *Aspergillus*
*conidia* [104]. Consequentially, this impairment of glycolysis, mediated by mammalian target of rapamycin (mTOR) and hypoxia-inducible factor 1 subunit alpha (HIF-1α), decreases macrophages’ conidicidal ability by lowering ROS concentration and inflammatory cytokines IL-1β, IL-6, IL-17A, TNF-α, and IFN-γ production [105]. 

*A. fumigatus* can be cleared by activating the complement system [106,107]. Tsai et al. further demonstrated that disruption of the arp1 gene leads to increased C3 deposition on the conidial cell surface [108]. Similarly, disrupting *alb1*/*pksP* that encodes polyketide synthase resulted in a significant increase in C3 binding on conidial surfaces, with an expected increase in phagocytosis by neutrophils and a decrease in virulence [89,109]. Direct binding of C3 fragments in normal human serum has been shown with *A. niger* melanin [76].

## 5. Other Melanotic Fungi and Their Interactions with the Immune System

*Fonseca* spp. are causative agents of chromoblastomycosis, and these fungi produce large quantities of melanin. Melanized *Fonsecaea monophora* and cell wall-containing extracted melanin significantly decrease the expression of inducible nitric oxide synthase gene and the production of nitric oxide and enhanced non-protective Th2 responses [110]. Macrophages infected with pigmented *F. monophora* enhanced the differential expression of genes related to immune responses, including the MAPK signaling pathway, demonstrating how melanization modifies pathogenesis [111]. As with *C. neoformans* and *A. niger* melanin, melanized *Fonsecaea pedrosoi* was quickly labeled with C3, C4, and C9 complement components [112].

Several endemic dimorphic pathogenic fungi produce melanin, including *Histoplasma capsulatum* [38], *Paracoccidoides* spp. [36], *Coccidioides immitis* [37], *Blastomyces dermatitidis* [15], and *Talaromyces marneffei* [113]. Melanin production is associated with pathogenesis in *Paracoccidioides* spp. through complex processes that extend beyond pigment production. Different *Paracoccidioides* species resist phagocytosis of yeast cells by macrophages, and this effect is associated with the degree of melanization in each strain [114,115]. Melanized *P. brasiliensis* is also highly resistant to NO, ROS, hypochlorite, and H_2_O_2_ [116]. A recent proteomic analysis comparing melanized and non-melanized *P. brasiliensis* and *P. lutzii* revealed that melanization leads to an abundance of virulence-associated proteins, including heat-shock proteins, vesicular transport proteins, adhesins, superoxide dismutases, proteases, and phospholipases, which further underscores the complex mechanisms that occur along with melanin production to subvert the host [117]. As with cryptococcosis, melanin-binding antibodies are generated during murine as well as human infection with *P. brasiliensis* [114].

*T. marneffei* (formerly *Penicillium marneffei*) also utilizes melanin to avoid host defenses. Remarkably, the *T. marneffei* genome has 23 polyketide synthase genes and additional non-ribosomal polyketide synthase hybrid genes [118]. Mutants unable to form the polymer are more sensitive to antifungals, H_2_O_2_, and sodium dodecyl sulfate (SDS). Furthermore, melanized cells were significantly more resistant to phagocytosis and killing compared to melanin-deficient mutants [119]. A second study corroborated the capacity of melanized fungal cells to resist antifungals [120]. There is an interesting link to melanin in *T. marneffei* with tyrosine catabolism, which is essential for survival in the host cells [121].

Melanin is well described in *Sporothrix* spp., and the production of the polymer is closely linked to virulence [13,122,123]. Melanization of *S. globosa* leads to a reduction in antigen presentation by macrophages and facilitates the dissemination of the pathogen [124]. Melanin formation has been linked to increased dissemination in several *Sporothrix* species [125]. However, melanization of some *S. schenckii* strains may, instead, induce the formation of granuloma, which facilitates the survival of the yeast [126]. Melanin production in *Sporothrix* complex species is protective against diverse antifungal compounds [127,128].

Melanin is purported to be a major factor in the pathogenicity of *Mucorales* spp. A recent paper from the Ibrahim laboratory identified compounds that could selectively inhibit eumelanin production by *Rhizopus* sp. [129]. Moreover, the inhibition of melanin by one blocking compound, UOSC-2, led to the formation of spores that were more efficiently phagocytosed and killed in mouse lungs compared to melanized spores, and the albino spores were similarly more efficiently killed by human macrophages, verifying the importance of melanin in protection against host effector responses against this pigmented species.

## 6. Open Questions in Melanin Biology

The enigmatic polymer’s structure remains elusive. Although there are synthetic melanins and certain precursors may mimic aspects of melanin’s activities, the complexities of melanin’s structure suggest that these are poor surrogates and do not represent the different forms of melanin. In fact, melanin formation in vivo is markedly different from that generated in vitro [130,131]. Future dissection of the structural unit of melanin, as approached by Dr. Stark and Dr. Cassadevall, will continue to unravel the mystery of melanin’s form [132]. Moreover, differences between types of melanin may reveal important variables in their biology as well as which molecules are combined with their formation and scaffolding. The evolution of different melanin systems also remains mysterious—why do some fungi have a single mechanism for forming melanin and others different avenues for formation? How did this evolve differently in human pathogenic fungi versus environmental species, and did this contribute to their abilities to cause diseases?

Although the formation of melanin is mysterious, its degradation is similarly unknown. The capacity for fungal melanin to persist in tissues even after the pathogen has been killed can challenge the host’s responses to nearby viable cells. Additionally, the effect of fungal melanin on other immune and non-immune cells is still elusive and should be experimentally addressed. Melanin has been suggested to have applications that could impact diverse aspects of life, ranging from remediation of nuclear contamination [133] to space travel [134].

## 7. Summary

Melanin plays a major role in the pathogenesis of a wide variety of human pathogens such as *A. fumigatus*, *C. neoformans*, *A. fumigatus*, and dimorphic fungi such as *Histoplasma capsulatum*, *Paracoccidioides* spp., *Coccidioides immitis*, and *Blastomyces dermatitidis*. The polymer can be compared to a Swiss army knife as fungi use melanin to improve their survival in a variety of environments through an array of mechanisms. Since its discovery, our view on melanin has gradually evolved. Melanin not only helps fungi to resist radiation and enzymatic degradation in the environment, but it also helps fungi to wreak havoc in the human body. Melanin renders our antifungals less effective while preventing our innate immune system from actively clearing the fungal invaders by lowering ROS production and phagocytosis. In A. fumigatus, immune evasion also involves modulating dendritic cells. Though melanin seems like an all-powerful tool, our immune system has also created our own counter measures such as the MelLec (melanin-sensing C-type lectin receptor). Meanwhile, our macrophages are also tuned in to detect melanin. It is exciting to see what the future holds as we unravel more about the interplay between melanin and our immune system, which will likely lead to the discovery of new therapeutic targets to aid us in our ongoing struggle with pathogenic fungi.

## Figures and Tables

**Figure 1 jof-07-00264-f001:**
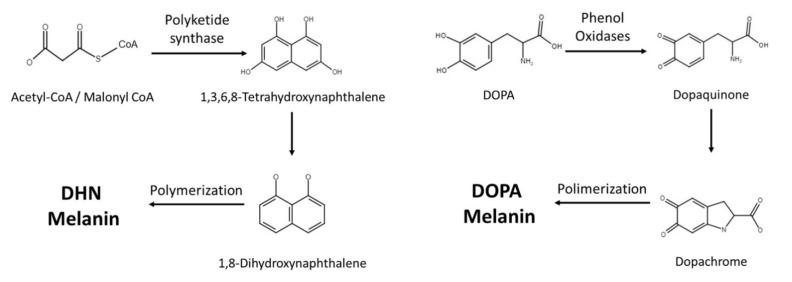
Current knowledge of melanin synthetic pathways in fungi.

**Figure 2 jof-07-00264-f002:**
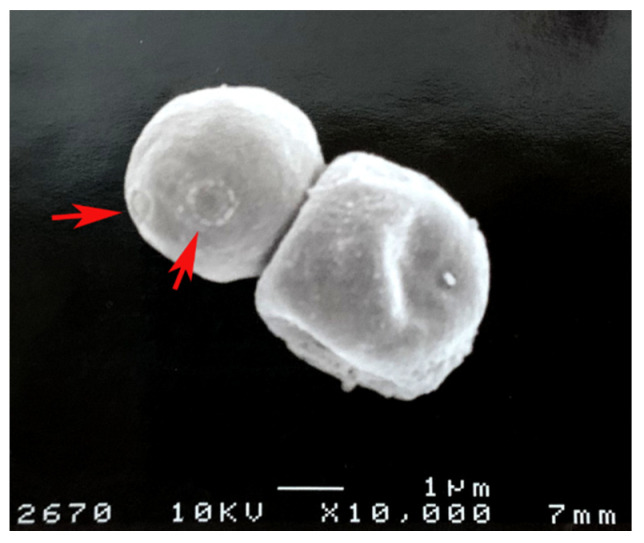
*Cryptococcus neoformans* melanin “ghosts” obtained from lungs of infected mice as described in [20]. The red arrows show bud scars on the melanin.

**Table 1 jof-07-00264-t001:** Representative fungi and the types of melanin they produce [12,13,14,15].

Species	Isolate Environment	Melanin Types
*Aspergillus fumigatus*	Clinical	DHN and pyo-melanin
*Aspergillus niger*	Industrial fermentation	DHN and L-DOPA
*Blastomyces dermatitidis*	Clinical	DHN
*Candida Albicans*	Clinical	L-DOPA
*Cryptococcus neoformans*	Clinical	L-DOPA
*Histoplasma capsulatum*	Clinical	DHN and L-DOPA
*Paracoccidioides brasiliensis*	Clinical	DHN and L-DOPA
*Fonsecaea monophora*	Clinical	DHN and L-DOPA
*Fonsecaea pedrosoi*	Clinical	DHN
*Sporothrix schenckii*	Clinical	DHN

Abbreviations: DHN is 1,8-dihydroxynaphthalene and L-DOPA is l-3,4-dihyroxyphenylalanine.

## Data Availability

Not applicable.

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
