# Peer review of "Fungal Melanin and the Mammalian Immune System"

_jof, 2021, doi:10.3390/jof7040264_

Round 1

Reviewer 1 Report

Overall, I think that this review is an excellent summary of the biological roles of DHN and L-DOPA melanin in fungal pathogenesis. It is comprehensive, well-sourced, and well-written. I have a couple of minor comments that might improve the document, but these should be considered suggestions and not by any means requirements.

  1. I have always been intrigued by the two melanin synthesis pathways, including their (presumed) redundancy in different species of fungi. A short section on the evolutionary aspects of melanin synthesis (or highlighting it as an understudied area) could be helpful for the reader.
  2. A figure outlining the two different synthesis pathways would be useful.

Author Response

Thank you for your thoughtful comments.

I have always been intrigued by the two melanin synthesis pathways, including their (presumed) redundancy in different species of fungi. A short section on the evolutionary aspects of melanin synthesis (or highlighting it as an understudied area) could be helpful for the reader.

  • This is an understudied area and there is limited information to address this. We have added this to a new section, #6, on questions for future consideration.

A figure outlining the two different synthesis pathways would be useful.

  • A new figure has been provided. (Personally, I find these unhelpful as the most important steps- the polymerization process- are completely unknown and just represented by an arrow...)

Reviewer 2 Report

Review: Fungal Melanin and the Mammalian Immune System This is a comprehensive literature review on the effects of melanin of medically important fungi to host immune response. The manuscript is well written and deserves publication with minor revisions. I would recommend for the authors to address the following points for improvement of the manuscript

  1. A figure on biosynthetic pathways of DOPA and DHN melanin with the intermediate enzymatic reactions is helpful for the reader
  2. A very important ref on a non-canonical secretory pathway regulating biosynthesis and trafficking of fungal melanin to the cell wall should be cited (ttps://pubmed.ncbi.nlm.nih.gov/26972005/)
  3. A brief description on the role of melanin in pathogenicity and intracellular persistence of Mucorales should be included
  4. The authors report challenges on the complex chemical structure of melanin pigments and the lack of data on crystallography of melanin polymers. They should emphasize the important functions of melanin related to the ability to inhibit immune effector functions are conserved in water soluble monomers (e.g., DHN). Accordingly, synthetic soluble melanin pigments, which mimic the functional aspects of fungal melanins, are available for immunological assays
  5. The authors should discuss on the common physicochemical properties of melanin pigments and how these properties are related to their biological functions. For example the anti-oxidant and metal binding properties of melanin deserve to be discussed
  6. A table or Box with open questions and future challenges on the field of melanin research would be desirable. a. For example, the effect of melanin on immune response has been focused mainly on macrophages. What is the mechanisms of interaction of melanin with other immune and non-immune cells? b. The lack of reliable antibodies and fluorescence probes to study melanin interaction with immune cells is a major obstacle in immunology research c. The mechanism of physiological degradation of melanin by the immune system is currently unknown. Is it enzymatic or non-enzymatic (e.g. ROS mediated) degradation? d. Possible interaction of melanin by products during degradation and sensing of these molecules by cytosolic sensors should be evaluated e. Precise mechanism of inhibition of calcium signaling by melanin (e.g., receptors and signaling molecules) deserves investigation f. The value of NMR in resolving structure of melanin needs to be discussed g. Briefly mention the importance of human melanin interaction with the immune systems

Author Response

We thank this reviewer for the insightful suggestions/comments

A figure on biosynthetic pathways of DOPA and DHN melanin with the intermediate enzymatic reactions is helpful for the reader

  • A figure has been added.

A very important ref on a non-canonical secretory pathway regulating biosynthesis and trafficking of fungal melanin to the cell wall should be cited (ttps://pubmed.ncbi.nlm.nih.gov/26972005/)

  • The reference has been included

A brief description on the role of melanin in pathogenicity and intracellular persistence of Mucorales should be included

  • Sentences on Mucorales have been added.

The authors report challenges on the complex chemical structure of melanin pigments and the lack of data on crystallography of melanin polymers. They should emphasize the important functions of melanin related to the ability to inhibit immune effector functions are conserved in water soluble monomers (e.g., DHN). Accordingly, synthetic soluble melanin pigments, which mimic the functional aspects of fungal melanins, are available for immunological assays

  • This is an interesting area, but these synthetics do not recapitulate all of the facets of fungal melanin. A brief discussion is included in the revision.

The authors should discuss on the common physicochemical properties of melanin pigments and how these properties are related to their biological functions. For example the anti-oxidant and metal binding properties of melanin deserve to be discussed

  • Additional text addressing this has been added. We and others have extensively reviewed these areas previously and the references are provided.

A table or Box with open questions and future challenges on the field of melanin research would be desirable. a. For example, the effect of melanin on immune response has been focused mainly on macrophages. What is the mechanisms of interaction of melanin with other immune and non-immune cells? b. The lack of reliable antibodies and fluorescence probes to study melanin interaction with immune cells is a major obstacle in immunology research c. The mechanism of physiological degradation of melanin by the immune system is currently unknown. Is it enzymatic or non-enzymatic (e.g. ROS mediated) degradation? d. Possible interaction of melanin by products during degradation and sensing of these molecules by cytosolic sensors should be evaluated e. Precise mechanism of inhibition of calcium signaling by melanin (e.g., receptors and signaling molecules) deserves investigation f. The value of NMR in resolving structure of melanin needs to be discussed g. Briefly mention the importance of human melanin interaction with the immune systems

  • This is a great suggestion and we have added a section that brings certain aspects of future areas for investigation into the discussion. 

Reviewer 3 Report

Congratulations for the excellent review on the role of melanin in fungal pathogenicity. The article has a very interesting content and is clear and well presented. However, I have one question and some suggestions on minor details that need to be corrected.

The question is about the sentence on page 4 line 132-134. The authors state that: C. neoformans cda1∆2∆3∆ mutants are avirulent in mice but, mice inoculated with this mutant generate a strong inflammatory response. What exactly do you mean by "avirulent", that this strain does not kill the animal? In my opinion it would be more appropriate to say that it is "less virulent" or that it has lost virulence compared to WT but is not completely avirulent.

Minor details:

About the use of italics:

  • Some parts of the text are written in italics and some others not without any apparent reason for it. Please, check the style and correct it accordingly.
  • All through the text, when citing authors followed by “et al”, as it corresponds to Latin, they have to be in italics (et al). The same for the names of fungi in Latin. They are sometimes in italics and sometimes not. Specifically, they are incorrect in: page 3, lines 85, 88, 90 and 91; page 5, line 175; page 6, line 237; page 7, line 314 and page 8, lines 338 and 339.

Typing errors:

  • In page 3 line 86 it seems that a word is missing at the end of the sentence: cryptococcal melanosomes explains how melanin is distributed in the cell wall and fungal (?)
  • In page 5, line 188 Laccase has to be laccase.
  • Page 5, line 214. Please review the beginning of the paragraph (In 1998…) it doesn’t fit with the rest of the sentence.
  • Page 6, line 261. You probably mean DOPA-melanin (instead of DOPE-melanin).

About references:

- All references are cited in the text, but they have to be renumbered as they jump from the 22 (page 3, line 75) to number 26 (line 78).

- Page 5 lines from 178 to 189. In this paragraph, two references are incorrectly cited:  in line 182 (Rosas et al., 2002) and line 185 (Goldman et al, 2000). Please, provide the correspondent number.

Author Response

We thank the reviewer for the comments and suggestions.

We agree with the issue of "avirulence" and have altered the text accordingly.

We have modified and corrected the text as detailed by the reviewer (under "minor")